Genome-wide identification of PP2A gene family in Camellia sinensis reveals the potential role of CsPP2A-TON2/FASS1 in abiotic stress

Bhattacharjee Surjit 1 2
Paul Abhirup 1
Jana Aradhana 1
Unnati G. Meher 1
R. Deepak 1
Miao Ye 3
Lu Honglin 4
Shen Guoxin 4 guoxin.shen@ttu.edu
Mishra Neelam 2 neelammishra@sju.edu.in
1 Bengaluru, Karnataka , India
2 Department of Botany, St. Joseph’s University , Bengaluru, Karnataka , India
3 Qingtian Forestry Technology Extended Station , Qingtian , China
4 Zhejiang Academy of Agricultural Sciences , Hangzhou , China
Uversky Vladimir
Electronic publication date: 2025 May 27
Publication date: 2025
Volume: 13
Electronic Location ID: e19431
Received 2024 Dec 5; Accepted 2025 Apr 16
Copyright: © 2025 Bhattacharjee et al.
Copyright year: 2025
Copyright holder: Bhattacharjee et al.
License: This is an open access article distributed under the terms of the Creative Commons Attribution License, which permits unrestricted use, distribution, reproduction and adaptation in any medium and for any purpose provided that it is properly attributed. For attribution, the original author(s), title, publication source (PeerJ) and either DOI or URL of the article must be cited.
License URL: https://creativecommons.org/licenses/by/4.0/

Keywords: Protein phosphatases, Genome-wide search, qRT-PCR, Stress-responsiveness, C. sinensis., Drought stress, Salt stress, PP2A gene family

Funding: Key Scientific and Technological Grant of Zhejiang (China) Breeding New Agricultural Varieties 2021C02072-5 This work was supported by Key Scientific and Technological Grant of Zhejiang (China) for Breeding New Agricultural Varieties (2021C02072-5). The funders had no role in study design, data collection and analysis, decision to publish, or preparation of the manuscript.

==============================
Background

Protein phosphatases (PP) play a crucial role in regulation of physiological response to various environmental stimuli in plants. Protein phosphatase 2A (PP2A) are a class of Ser/Thr protein phosphatases which are present widely across plant species and facilitate plant responses to biotic and abiotic stresses, and hormone treatment.

Methods

Using BLASTp, tea PP2A genes were found and their physicochemical characteristics (ExPASy), subcellular localization, and intron-exon structure (GSDS) were described. MEGA software was used to examine phylogenetic relationships, SMART online tool to analyze conserved domains, and PlantCARE tool to analyze cis-elements. The Heatmapper online tool was used to visualize expression profiles across plant tissues and stress conditions. Under controlled conditions, 2-year-old Camellia sinensis plants were exposed to methyl jasmonate (MeJA) stress treatments, cold drought, and salt stresses and qRT-PCR (2−ΔΔCt technique, CsACTIN as internal control) was used to validate the results.

Results

We found 11 PP2A genes in C. sinensis (CsPP2A) through a genome-wide search wherein Arabidopsis thaliana was taken as the reference genome. Further, a systematic study was conducted wherein physicochemical properties, phylogeny, gene structure and gene expression were explored. Analysis of cis-elements, gene ontology, and expression patterns of CsPP2A genes under various stresses, combined with prior research on the PP2A gene family in other plants, suggests that the PP2A family plays a role in regulating phytohormone (auxin, salicylic acid, gibberellin, abscisic acid, MeJA) responses during stress in C. sinensis. The real time PCR also confirmed the potential role of (TEA018948.1) CsPP2AB9/CsPP2A-TON2/FASS1 gene in abiotic stress responses.

Conclusion

This study offers potential goals for further in-depth investigation and functional analysis, alongside enhancing our understanding of the regulatory network of PP2A genes in C. sinensis.

Introduction

Camellia sinensis (tea plant) is a widely grown crop in tropical and subtropical regions across the globe with its leaves being consumed globally as a popular non-alcoholic beverage. However, the tea plant is highly susceptible to various abiotic stress conditions namely drought, salt and cold stresses, which significantly affect its growth and development (Xie et al., 2019; Wan et al., 2018; Li et al., 2018). These stresses lead to reduced crop yield, threatening both quality and quantity of tea production. Therefore, it is crucial to identify and characterize genes in the tea plant which enhances its abiotic stress responsiveness.

Fundamental mechanisms of regulation in biology involves controlled phosphorylation and dephosphorylation of proteins (Hunter, 1995; Máthé et al., 2023). Reversible phosphorylation of proteins catalysed by protein phosphatases (PPs) and protein kinases, as part of protein modification, plays a role in regulation of various physiological responses in plants (Wang, Liang & Lu, 2001; Kong et al., 2013). PPs reverse the phosphorylation by removing phosphate groups and these enzymes can be classified into three categories: Protein Tyr phosphatases (PTPs), Ser/Thr phosphatases (STPs), and dual specificity phosphatases (DSPTPs) (Cohen, 1989; Li et al., 2014; Zhang et al., 2022). Specifically, PP2As (Protein phosphatase 2A) are a class of Ser/Thr protein phosphatases that have a crucial role in controlling phosphorylation of proteins by eliminating their phosphate groups (Perrotti & Neviani, 2013; Creighton et al., 2017).

The three subunits, namely subunit A (structural), subunit B (regulatory) and subunit C (catalytic) together form the PP2A holoenzyme (Kataya et al., 2015; Booker & DeLong, 2017). Subunit A is made up of a series of conserved alpha helices and serves as a scaffold for the subunits B and C to bind (Chen et al., 2014). Moreover, substrate specificity is determined by subunit B whereas the dimerization of subunit A and C results into its active conformation, and caters the interactions with subunit B (Chen et al., 2014). In previous studies, the B subunits are further sub classified into B, B′, B″, and B55 (subclass: TON2) on the basis of their structural characteristics (Tsugama et al., 2018; Lillo et al., 2014). This assortment of PP2A subunits contribute to the essential role of PP2As in biological processes (Durian et al., 2016).

Mutant studies suggests the involvement of certain catalytic subunits of AtPP2A in ABA (abscisic acid) signaling and auxin fluxes (Garbers et al., 1996; Michniewicz et al., 2007; Pernas et al., 2007; Ballesteros et al., 2013). It has been observed that PP2As in Arabidopsis regulate certain enzymes involved in important processes like nitrogen assimilation and their regulation is strongly influenced by light (Douglas et al., 1997; Huber, MacKintosh & Kaiser, 2002; Heidari et al., 2011). Moreover, there are certain microtubule-associated PP2A proteins that are crucial for controlling the dynamics and organization of microtubules (cortical and mitotic) in plants (Janssens & Goris, 2001; Camilleri et al., 2002; Yoon, Ahn & Pai, 2018; Vavrdová, Samaj & Komis, 2019). Owing to the importance of PP2As in plants, these genes were previously studied in plants such as rice and rubber (Yu et al., 2003; Chao et al., 2020). In rice, few PP2A-A subfamily genes were found to be upregulated during high salinity and drought stress conditions (Yu et al., 2003). However, PP2A is not explored in tea which is an economically important crop with several health benefits to human population (Mondal et al., 2004). Furthermore, PP2A is considered as an anti-stress protein and therefore, it is essential to characterize and study these genes in tea (Deng et al., 2020).

In the present study, we carried out genome wide analysis of PP2A in C. sinensis genome using the Arabidopsis thaliana genome as the reference. We identified 11 PP2A genes and classified them into different subfamilies based on the phylogenetic analysis. Further, their structures, Gene Ontology, protein interaction network, spatial expression as well as expression under abiotic stress conditions were analyzed. Additionally, real time PCR data revealed the involvement of CsPP2AB9/CsPP2A-TON2/FASS1 gene in regulation of drought, salt and cold stress in C. sinensis.

Materials and Methods

Plant growth and stress treatments

The tea plants (C. sinensis var. sinensis cv shuchazao) were grown at the Huzhou Academy of Agricultural Sciences, China. Two-year-old tea plants from the same seedbed (free of pests and diseases and about 20 cm in height) were selected and transplanted into pots (Danish peat soil). The artificial climate chamber growth conditions were set as follows: a temperature of 22 ± 2 °C, humidity of 65%, untreated plants as control (CK), simulated drought (25% PEG 6000), and NaCl (200 mM for 72 h), 25% polyethylene glycol for 72 h (PEG 6000), methyl jasmonate for 48 h (MeJA) and cold stress of 4 °C for 7 days. The seedlings were then placed in a growth chamber at 23 °C with a 16 h photoperiod of 10,000 lux light intensity followed by 8 h of darkness. Three to four leaves were collected after stress treatments at mentioned time points, rapidly frozen in liquid nitrogen, and stored at −80 °C until further use. Each experiment was performed with three biological replicates.

Identification and characterization of PP2A genes in tea

The Arabidopsis (Arabidopsis thaliana) and rubber (Hevea brasiliensis) PP2A gene and protein sequences for both PP2A-A and PP2A-B (B′, B″, B55 and TON2) subfamilies were retrieved from The Arabidopsis Information Resource (TAIR) (Berardini et al., 2015) (https://www.arabidopsis.org/) and rubber genome database (RIKEN) (Lau et al., 2016) (http://rubber.riken.jp/home.html), respectively. These protein sequences were then utilized as query sequences to search for the PP2A genes in C. sinensis from Tea Plant Information Archive (TPIA) (Xia et al., 2019) (http://tpia.teaplant.org/) by employing the BLASTp algorithm. A threshold identity percentage of 50% and e-value of 1e−5 were selected as the BLASTp parameters. Furthermore, for characterization of the identified PP2A genes in tea, the physicochemical properties were analysed using the ExPASy ProtParam tool (Gasteiger et al., 2005) (https://expasy.org/). DeepLoc—2.0, CELLO (subcellular localization predictor) WoLF PSORT (Advanced Protein Subcellular Localization Prediction Tool), and MULocDeep (Thumuluri et al., 2022;Yu, Lin & Hwang, 2004; Yu et al., 2006; Horton et al., 2007; Yu et al., 2010; Jiang et al., 2021; Jiang et al., 2023), (http://cello.life.nctu.edu.tw/) (https://www.mu-loc.org/) (https://services.healthtech.dtu.dk/services/DeepLoc-2.0/) (https://www.genscript.com/wolf-psort.html) servers were used to predict the sub-cellular locations of the putative PP2A genes in tea. In addition, the identified PP2A genes were screened for the presence of transmembrane helices using TMHMM server v2.0 (Sonnhammer, Von Heijne & Krogh, 1998) (https://busca.biocomp.unibo.it/bacello/).

Phylogenetic analysis of the tea PP2A genes

PP2A peptide sequences for both A and B subfamilies retrieved from Arabidopsis and rubber, along with the identified tea peptide sequences were used to analyze their phylogenetic relationships. The phylogenetic trees were constructed using the MEGA 7.0.14 software (Kumar, Stecher & Tamura, 2016) (https://www.megasoftware.net/) by employing the Neighbor Joining (NJ) algorithm. All parameters were kept at default settings and number of bootstrap replicates was set to 1,000.

Analysis of CsPP2A genes structure

The putative PP2A genes for both A and B subfamilies in tea were further analyzed to check their intron-exon distribution patterns. The gene and CDS sequences of tea PP2A genes were uploaded to the Gene Structure Display Server (GSDS 2.0) tool (Hu et al., 2015) (http://gsds.cbi.pku.edu.cn/) to check the presence of the untranslated Regions (UTRs) and intron-exon segments.

Similarity index and domains of the tea PP2A proteins

The tea PP2A for both the A and B subfamilies were checked for their similarity indexes against their homologous counterparts in A. thaliana. The peptide sequences were aligned using MUSCLE web tool (Robert, 2004) (https://www.ebi.ac.uk/Tools/msa/muscle/) with default parameters and the respective similarity index between the genes of the individual subfamilies were calculated. Peptides sequences of Arabidopsis and tea PP2A genes were used to identify the domains via SMART online tool (Letunic, Khedkar & Bork, 2020) (http://smart.embl-heidelberg.de/).

Genomic distribution map and analysis of cis-regulatory elements of tea PP2A genes

The tea PP2A genes were mapped onto their respective genomic scaffolds using the MapGene2Chromosome web v2 (MG2C) server (Chao et al., 2015) (http://mg2c.iask.in/mg2c_v2.1/). The genes had to be mapped on their scaffolds owing to the fact that TPIA database has inadequate genome assembly data. Moreover, tea PP2A upstream sequences stretching 2,000 bp were retrieved from TPIA database and scanned for the presence of cis-acting regulatory elements (CAREs) using the PlantCARE program (Rombauts et al., 1999; Lescot et al., 2002) (http://bioinformatics.psb.ugent.be/webtools/plantcare/html/).

Gene Ontology analysis and functional interaction network of tea PP2A genes

The Gene Ontology (GO) identifiers of all the tea PP2A genes for both the A and B subfamilies were sourced from TPIA database. These GO identifiers were then analysed using QuickGO server (https://www.ebi.ac.uk/QuickGO/). STRING online tool (Szklarczyk et al., 2019) (https://string-db.org/) was used to find the functional network of interacting homologous genes among tea and Arabidopsis PP2A genes.

Structural comparison of tea and Arabidopsis PP2A proteins

The homology modelling of PP2A proteins for tea and Arabidopsis plants was carried out using ExPASy Swiss-model tool (https://swissmodel.expasy.org/). The predicted protein structures were evaluated using ProSA-web server (https://prosa.services.came.sbg.ac.at/prosa.php). PyMOL software (https://pymol.org/2/) was used to analyse the structural similarities of predicted protein models.

Expression profiles of tea PP2A genes using transcriptome data

The tea PP2A gene expression profiles for both A and B subfamilies were procured from the TPIA database. The database houses expression data for eight tissues namely flower, apical bud, young leaf, mature leaf, old leaf, root, and stem. Expression levels in eight different tissues have been experimentally determined (Wei et al., 2018). In the present study, expression analysis was carried out for root, stem, mature leaf, old leaf, young leaf, apical bud, flower, and fruit (Table S6). Furthermore, the gene expression patterns of these PP2A genes under various abiotic stress (cold, drought, salt) conditions and under methyl jasmonate (MeJA) treatment were also analysed. According to the cold acclimated data in the TPIA database, five stages of expression have been studied: (i) Non-acclimated: 25~20 °C (CK—control), (ii) Fully acclimated: 10 °C for 6 h (CA 1-6h), (iii) Cold treatment: 10~4 °C for 7 days (CA 1-7d), (iv) Cold treatment at 4~0 °C for 7 days (CA 2-7d), and (v) Recovery condition: 25~20 °C for 7 days (DA-7d) (Xia et al., 2019; Wang et al., 2013) (Table S7).

Drought stress simulation included four stages under 25% polyethylene glycol (PEG) treatment for (i) 0 h (control), (ii) 24 h, (iii) 48 h, and (iv) 72 h (Zhang et al., 2017) (Table S8). Salinity stress was simulated in tea plant using 200 mM NaCl and the response was studied based on increasing exposure time: 0 h (control), 24 h (NaCl-N-24h), 48 h and 72 h (Zhang et al., 2017) (Table S9). Effect of MeJA treatment on tea PP2A genes was studied by exposing the plant parts of tea to MeJA solution for increasing time intervals: 0, 12, 24 and 48 h (Xia et al., 2019; Shi et al., 2015) (Table S10). Heatmapper online server (Babicki et al., 2016) (http://www.heatmapper.ca/) was used to generate the corresponding heat maps for the expression graphs.

Quantitative real time polymerase chain reaction analysis

The primers for of PP2A-B subfamily genes required for quantitative real time polymerase chain reaction (qRT-PCR) were designed through Primer Blast tool (Ye et al., 2012) (https://www.ncbi.nlm.nih.gov/tools/primer-blast/). Total RNA (1 µg) of each sample was treated with RNase-free DNase I (Takara, Shiga, Japan). The extracted RNA showed purity 260/280 ratio in the range of 2.0–2.2 and yield was approximately 10–20 µg for all samples. The extracted RNA was either utilized immediately or stored in −80 °C until further use. The RNA was used to synthesize the first strand of cDNA using the PrimeScript RT reagent kit (Takara, Shiga, Japan). The cDNA was used as a template in the qRT-PCR using SYBR Green I Master Mix (Roche Diagnostics, Indianapolis, IN, USA) on an ABI 7500 system (Applied Biosystems, Waltham, MA, USA). The study was conducted with two biological and three technical replicates. The qRT-PCR condition was set as: 95 °C for 30 s, followed by 40 cycles of 95 °C for 5 s, 60 °C for 30 s, and 1 cycle of 60 °C for 1 min. The CsACTIN gene being a constitutive/housekeeping gene was selected as an internal control. The relative expression levels of genes were calculated using 2−ΔΔCt method. To examine the effects of different environmental/abiotic stresses, expression profiles of the PP2A genes were studied under the abiotic stresses namely salt, drought and cold as well as one phytohormone stress. The tea plants were subjected to various abiotic stress conditions such cold stress (0–4 °C for 7 days), drought stress (25% PEG for 72 h), salt stress (200 mM NaCl for 72 h) and Methyl Jasmonate treatment (MeJA for 48 h). The duration of stress conditions was determined based on the experiments conducted by Xia et al. (2019), which examined the expression of the CsPP2A genes under various abiotic stresses. qRT-PCR analysis was performed on samples subjected to both control and abiotic stress conditions, including cold stress, drought stress, salt stress, and Methyl Jasmonate (MeJA) treatment. The gene-specific primers used for qRT-PCR are listed in Table S11.

Data analysis

Student t-test considering one tailed unequal variance was performed to compare the differences between treatments. Statistical analysis was performed using Microsoft Office Excel 2007. **p ≤ 0.001 is significance level shown in the data.

Results

Identification and characterization of PP2A genes in tea

Among the 11 PP2A genes identified in tea, two genes belong to the PP2A-A subfamily whereas PP2A-B55, PP2A-B′, PP2A- B″, and PP2A-TON2 subfamilies contain 1, 5, 2 and 1 genes, respectively (Table S1).

To characterize the gene family, it is essential to analyze physicochemical properties such as grand average of hydropathy (GRAVY) index, aliphatic index, and instability index. The physicochemical properties of these genes were inferred via the ProtParam tool (Gasteiger et al., 2005) in ExPASy (Table 1). The CsPP2A gene lengths (bp) found to extend from 5,156 bp to 18,430 bp, and protein lengths (aa) between 433–1,268 aa, were downloaded from TPIA database (Table 1). The molecular weight of the PP2A proteins ranged from 49,400 to 143,000 kDa, with TEA019300.1 (CsPP2A-B3/CsPP2AB′β) being the heaviest at 142,997.68 kDa and TEA018948.1 (CsPP2A-B9/CsPP2A-TON2/FASS1) the lightest at 49,407.77 kDa. The theoretical pI values show that CsPP2A proteins (eight out of 11) are generally acidic (pI < 7). However, there are three genes from PP2A-B′ subfamily namely TEA015045.1 (CsPP2A-B4/CsPP2AB′γ), TEA009324.1 (CsPP2A-B5/CsPP2AB′δ) and TEA000454.1 (CsPP2A-B6/CsPP2AB′ε), that are basic (pI > 7). The GRAVY index value was negative for most of the tea PP2As (nine proteins) whereas the remaining two CsPP2As showed positive GRAVY index. GRAVY index represents the hydrophobicity of a whole molecule wherein negative values suggest hydrophilicity while positive values suggest a hydrophobic nature (Walker, 2005). Therefore, it was found that most of the CsPP2As are hydrophilic in nature whereas two genes are hydrophobic All the CsPP2A proteins displayed an instability index above 40 except CsPP2A-B4 with a score of 39.86, which is extremely close to 40. The instability index provides a measure of protein stability in vitro wherein values near 40 or above represents stable nature of the protein (Walker, 2005; Wang et al., 2018). The PP2A proteins in tea have aliphatic indices ranging from 70.86 (CsPP2A-B1/CsPP2AB55α) to 115.17 (CsPP2A-A1). Among all the PP2A subfamilies in tea, PP2A-A subfamily has highest aliphatic index. Aliphatic index refers to the volume occupied by side chains that are aliphatic in nature (Walker, 2005; Ikai, 1980). Moreover, aliphatic side chains might have a significant role in determining the thermostable nature of a protein depending on the percentage of individual aliphatic amino acid present (Ikai, 1980; Lu, Wang & Huang, 1998). Similarly, proteins with greater number of hydrophobic amino acid residues also exhibit higher thermostability as observed in the study of Mozo-Villarías, Cedano & Querol (2006).

Table 1 Detailed analysis of the physicochemical properties and sequence characteristics of all the putative PP2A (A and B subfamily) genes in C. sinensis.

Gene ID	Locus position	Gene length (bp)	Protein length (aa)	Mol. Wt. (KD)	pI value	No. of negative residues	No. of positive residues	GRAVY index	Instability index	Aliphatic index	Subcellular localization (Wolf PSORT)	Subcellular localization (MULocDeep)	Subcellular localization (Cello)	Subcellular localization (DeepLoc 2.0)	
PP2A-A Subfamily	
TEA002042.1 (CsPP2A-A1)	Scaffold 3595: 2,651,300−
2,682,014-	30,714	513	57,290.61	4.79	72	46	0.166	49.80	115.17	Cytoplasm	Cytoplasmic	Cytoplasmic	Cytoplasm	
TEA011483.1 (CsPP2A-A2)	Scaffold 3538: 346,370−
359,948+	13,578	513	57,102.18	4.71	72	44	0.157	50.44	114.02	Chloroplast/cytoplasmic	Cytoplasmic	Cytoplasmic	Cytoplasm	
PP2A-B_B55 Subfamily	
TEA015525.1 (CsPP2A-B1/CsPP2AB55α)	Scaffold 2145: 3,114,501-
3,128,741+	14,240	522	57,870.77	5.91	65	56	−0.434	49.40	70.86	Cytoskeleton/nuclear	Cytoplasmic	Nuclear	Nucleus/cytoplasm	
PP2A-B’ Subfamily	
TEA021355.1 (CsPP2A-B2/CsPP2AB’α)	Scaffold 2153: 1,496,800−
1,503,585+	6,785	487	56,439.37	5.83	70	59	−0.179	47.92	99.47	Mitochondria/nucleus/cytoplasm	Cytoplasmic	Cytoplasmic	Cytoplasm/nucleus	
TEA019300.1 (CsPP2A-B3/CsPP2AB’β)	Scaffold 2295: 666,933−
685,363−	18,430	1,268	142,997.68	6.22	161	145	−0.441	47.94	83.82	Chloroplast/nucleus	Mitochondria	Nuclear	Cytoplasm/nucleus	
TEA015045.1 (CsPP2A-B4/CsPP2AB′γ)	Scaffold 95: 500,134−
510,636+	10,502	570	64,836.39	8.76	63	70	−0.260	39.86	91.16	Mitochondria/nucleus/cytoplasm	Cytoplasmic/mitochondria	Nuclear	Cytoplasm/nucleus	
TEA009324.1 (CsPP2A-B5/CsPP2AB′δ)	Scaffold 3744: 183,567−
188,723−	5,156	505	57,398.25	8.45	55	59	−0.132	49.93	92.46	Chloroplast/mitochondria/cytoplasm	Nucleus	Cytoplasm/Plasma membrane	Cytoplasm/nucleus	
TEA000454.1 (CsPP2A-B6/CsPP2AB′ε)	Scaffold 267: 913,511−
924,709+	11,198	583	66,150.02	7.56	69	70	−0.186	42.84	88.30	Cytoplasm/chloroplast	Cytoplasmic	Nuclear	Cytoplasm/nucleus	
PP2A-B” Subfamily	
TEA000364.1 (CsPP2A-B7/CsPP2AB′′α)	Scaffold 890: 308,384−
325,692+	17,308	534	61,391.83	4.90	80	55	−0.366	43.86	80.00	Nucleus/cytoplasm	Cytoplasmic	Cytoplasmic	Cytoplasm/nucleus	
TEA021728.1 (CsPP2A-B8/CsPP2AB′′β)	Scaffold 621: 1,132,090−
1,145,074−	12,984	718	81,336.44	5.05	93	66	−0.256	47.21	86.11	Nucleus/cytoplasm	Nucleus	Cytoplasmic	Cytoplasm/nucleus	
PP2A-B-TON2 Subfamily	
TEA018948.1 (CsPP2A-B9/CsPP2A-TON2/FASS1)	Scaffold 10057: 176,498−
187,973−	11,475	433	49,407.77	4.86	78	50	−0.461	46.24	80.28	Nucleus/cytoplasm	Nucleus	Cytoplasmic	Cytoplasm/nucleus	

The PP2A proteins of C. sinensis were predicted to be localized in cytosol, nucleus, chloroplast, mitochondria and plasma membrane. The subcellular localization pattern of CsPP2A-B subunit proteins is consistent with the localization pattern of PP2A-B subunit proteins in Arabidopsis being found in the cytosol, nucleus, and plasma membrane (Máthé et al., 2023). Furthermore, transmembrane analysis reveals the absence of transmembrane helices in all CsPP2A proteins (Fig. S1).

Phylogenetic relationship of the tea PP2A genes

This phylogenetic tree was constructed using protein sequences from A. thaliana (20 sequences), H. brasiliensis (28 sequences) and C. sinensis (11 sequences). The NJ tree constructed for phylogenetic analysis resulted into five different clades (based on bootstrap replicates) and thus, we have classified the CsPP2A gene family into five subfamilies based on A (scaffolding) subunit and B (regulatory) subunit, namely PP2A-A, PP2A-B55, PP2A-B″, PP2A-B′ and PP2A-TON2 (Fig. 1). Although PP2A-TON2 was sub classified under PP2A-B55 subfamily in other plants, we have considered it as a separate subfamily in tea due to its clade emerging separately from that of CsPP2A-B55. This analysis facilitates the classification of PP2A homologs in C. sinensis.

Figure 1 Phylogenetic tree of PP2A genes from Arabidopsis thaliana (green), Hevea brasiliensis (black) and Camellia sinensis (red).

The PP2A protein sequences were aligned using MUSCLE, and the phylogenetic tree was constructed using MEGA 7.0.14 by the Neighbor-Joining (NJ) method with default parameters and 1,000 bootstrap replicates. The tree is divided into five major clades of PP2A genes, based on their subfamilies PP2A-B′ subfamily, PP2A-TON2 subfamily, PP2A-A subfamily, PP2A-B55 subfamily and PP2A-B″ subfamily.

Intron—exon architecture of the PP2A genes in tea

Owing to its importance in evolution, the intron-exon architecture was studied based on the phylogenetic relationship between CsPP2A and AtPP2A genes. Analysis of gene structures reveals that non-coding sequences are abundant. This genome analysis related to the abundance of non-coding sequences alludes to genome complexity (Taft, Pheasant & Mattick, 2007; Goyal et al., 2018; Chatterjee et al., 2020; Paul et al., 2021). The intron-exon organization of the 11 CsPP2A genes revealed the wide variation in the number of introns/exons across the subfamilies. All the subfamilies of CsPP2A contain genes with UTRs at both N-terminal and C-terminal ends except two genes in the PP2A-B′ subfamily (CsPP2AB’β and CsPP2AB′ε containing a UTR at their C-terminal end) and two genes in the PP2A-B″ subfamily (CsPP2AB′′β and AtPP2AB′′α containing a UTR at their N-terminal end) (Fig. 2). Further, it can be observed that although the length and arrangement of introns/exons among most of the genes of same subfamily varied, their numbers remained similar. For instance, both the tea PP2A-A subfamily genes (CsPP2A-A1 and CsPP2A-A2) contained nine introns and 10 exons whereas the count of exons in CsPP2B′ genes of PP2A-B′ subfamily ranged between 2–4 (intron count varied from 1–3), except CsPP2AB′β (16 exons and 15 introns) with maximum number of exons and introns among all the CsPP2A genes. CsPP2AB″α and CsPP2AB″β genes of the PP2A-B″ subfamily possessed 12 exons/11 introns and 15 exons/14 introns respectively. On the other hand, 14 exons/13 introns were observed in the CsPP2AB55α gene of PP2A_B55 subfamily and 11 exons/10 introns in the CsPP2A-B9/CsPP2A-TON2/FASS1 gene of PP2A-TON2 subfamily. Additionally, it was observed that PP2A genes of Arabidopsis and Tea that are positioned within the same clade exhibit highly comparable structural features, including similar numbers of introns, exons, and UTRs. For instance, AtPP2AB55B and CsPP2AB55α both possess 14 exons, 13 introns, and two UTRs, while AtPP2A-TON2/FASS1 and CsPP2A-TON2/FASS1 contain 12 exons/11 introns and 11 exons/10 introns, respectively. These similarities in gene structure support the classification of CsPP2A genes into five distinct subfamilies, while also suggesting potential structural diversity due to slight variations in exon/intron counts.

Figure 2 The intron/exon structures of PP2A genes in tea plant.

Gene Structure Display Server 2.0 was used to draw gene structure maps. Red boxes represent exons, black boxes represent the UTRs, and black lines represent introns. The gene length can be estimated by using the scale (in kb) given at the bottom. The clades divided based on their subfamilies.

Similarity index and domains of the tea PP2A proteins

Similarity index helps to analyze the identity of amino acid sequences which in turn supports the sequence conservation among PP2A genes in tea. In A subfamily, tea PP2A showed high amino acid sequence identities with Arabidopsis PP2A (CsPP2A-A1/AtPP2A-A1 = 85.96%, CsPP2A-A1/AtPP2A-A3 = 91.23%, CsPP2A-A1/AtPP2A-A2 = 94.54% CsPP2A-A2/AtPP2A-A1 = 84.99%, CsPP2A-A2/AtPP2A-A2 = 92.4%, CsPP2A-A2/AtPP2A-A3 = 89.28%). Amongst the B-subfamily similar results were observed where-in CsPP2AB55α which grouped with AtPP2A-B55α and AtPP2A-B55β exhibited 79.84% and 84.46% sequence identity with both the proteins respectively, CsPP2AB′α grouping with AtPP2AB′α, AtPP2AB′β and AtPP2AB′ε displayed 69.28%, 70.87% and 62.95% respectively. Similarly, CsPP2AB′β which grouped with AtPP2AB′η and AtPP2AB′θ, CsPP2AB′γ grouping with AtPP2AB′γ and AtPP2AB′ζ, CsPP2AB′δ and CsPP2AB′ε grouping with AtPP2AB′κ, CsPP2AB′′α grouped with AtPP2AB′′α and AtPP2AB′′β, CsPP2AB′′β grouped with AtPP2AB′′α, AtPP2AB′′β, AtPP2AB′′γ, AtPP2AB′′δ, AtPP2AB′′ε and CsPP2A-TON2/FASS1 grouping with AtPP2A-TON2/FASS1 displayed sequence similarity greater than 50% (Fig. 3). This analysis highlights the sequential similarity between Arabidopsis and tea PP2A proteins validating the phylogenetic classification of CsPP2A proteins.

Figure 3 Similarity index.

Similarity matrix between the PP2A genes of Arabidopsis and tea was constructed using MUSCLE tool. The lightest regions represent minimum percentage identity and darkest shades represent maximum identity between the sequences of PP2A in A subfamily (3A) and B subfamily (3B). The intensity of the color increases with increasing values.

Using the SMART online tool, a correlative investigation of conserved protein domains in Arabidopsis and tea was conducted to assess the structural profile and diversity of these proteins. A B subunit PP2A proteins of both Arabidopsis and tea displayed similar domains (Fig. 4). A few low complexity regions were observed in certain members of the PP2A-B subfamilies (PP2A-B_B55, PP2A-B′, PP2A-B″ and PP2A-TON2) whereas these regions did not appear in the PP2A-A subfamily. On the other hand, each clade showed at least one domain common to all their members. All the PP2A-A subfamily proteins had 1–2 heat domains. In addition, an internal repeat was present in this subfamily members except in AtPP2A-A3. The PP2A-B′ subfamily uniformly exhibited 1–2 B56 domains and CsPP2A-B3/CsPP2AB′β additionally had a HAUS_N domain. This HAUS augmin-like complex has a role in maintaining centrosome integrity, assembly of mitotic spindle and cytokinesis. It is also involved in microtubule polymerization via an interaction between its subunit 6 and NEED-1-gamma-tubulin complex. WD-40 repeats or beta-transducin repeats (5–7) were observed in the PP2A-B_B55 subfamily (AtPP2A-B55α, AtPP2A-B55β and CsPP2AB55α). These repeats are implicated in diverse functions including signal transduction, assembly of protein complexes and transcriptional control. Moreover, several proteins with WD-40 repeats in Arabidopsis spp. play a major role in plant growth developmental events. All the members of Arabidopsis and tea PP2A-B″ and PP2A-TON2 subfamilies exhibited 2 or 3 EF-hand domains (EFh or EF-hand_5 or EF-hand_7). These are calcium-binding motifs that undergo a conformational alteration when calcium ions bind. EF-hands occur at least in pairs wherein each motif has a 12-residue loop with a 12-residue alpha helix flanked on its either side. Further, a unique PDB domain: 2VG6G|A possessing oxidoreductase function was found in CsPP2AB″β of PP2A-B″ subfamily. Therefore, the characteristic domains of each subfamily with specific function indicates a correlation between their evolutionary relationship and functional characteristics.

Figure 4 Domain analysis of PP2A proteins in tea plant.

SMART online tool was used to identify the domains.

Genomic distribution map and analysis of cis-regulatory elements of tea PP2A genes

To get an insight on the gene distribution pattern, CsPP2A genes were mapped onto the scaffolds. Each CsPP2A gene was found to be located on a separate scaffold (Fig. S2). Therefore, 11 scaffolds with 11 different CsPP2A genes were obtained via the MapGene2Chromosome web v2 (MG2C) server.

Initially, the promoter sequences, 2,000 bp upstream of the initiation codon “ATG” were obtained from TPIA database. Further, these sequences were used as input in the PlantCARE database to identify the cis-acting regulatory elements (CAREs). This analysis revealed the presence of 45 CAREs across the PP2A genes in tea. Based on their specific roles in the plant, the CAREs were grouped under 22 different categories and represented in a pie chart (Fig. 5). Moreover, the sequence length of the identified cis-acting regulatory elements was observed to be ranging between 5 and 14 bp (Fig. 5). Among these, majority of the CAREs had a sequence length of either 13 or 10 bp (Fig. 5). In total, 45 CAREs were detected and these were distributed across the 11 CsPP2A genes. Our study of CAREs in tea revealed multiple elements that seem to play a role in light responsiveness, stress responses, and hormonal regulation (Table S2). About 20 CAREs play a role in light responsiveness (Fig. 5). Moreover, HD-zip 3 and Box III were elements functioning in protein-binding site. CAREs involved in response to hormonal treatment included ABRE (Abscisic acid responsiveness), TGA-element (Auxin response element), P-box/GARE-motif (Gibberellin-responsive Element), TCA element (Salicylic acid responsiveness), and TGACG-motif/CGTCA-motif (MeJA responsiveness). GARE-motif was identified to be involved in defense and stress responsiveness. Further, CAREs in tea found related to abiotic stress responses such as low-temperature responsiveness and drought-inducibility were LTR and MBS respectively. MSA-like element, AACA-motif, CAT-box, GCN4-motif, ARE, AT-rich sequence, and O2-site are few other identified cis-acting regulatory elements. Analysis of cis-acting regulatory elements (CAREs) helps comprehending the effect of distinct plant responses to stress conditions and environmental cues on growth regulation (Akram et al., 2020).

Figure 5 Analysis of cis-acting elements identified from the PP2A genes in tea plant.

PlantCARE database was utilized to identify all the cis-acting elements of PP2A genes. Pie chart showing the frequency of different cis-acting elements based on their specific biological activities. Histogram showing the frequency of different sequence lengths of the cis-acting elements.

Gene ontology (GO) analysis and Functional Interaction network of tea PP2A genes

GO analysis was performed using text mining, to predict PP2A functions in C. sinensis. Our study revealed 17 GO terms in tea PP2A genes (Table S3) obtained from TPIA database. It includes three categories namely biological process (6 GO terms), cellular component (7 GO terms), and molecular function (4 GO terms) (Fig. 6). The GO terms were analysed using QuickGO server (Binns et al., 2009). Our analysis in tea revealed three genes (CsPP2A-B1/CsPP2A-B4/CsPP2A-B5) as a part of protein phosphatase type 2A complex (GO:0000159) (Rossio et al., 2013) and two genes (CsPP2A-B4/CsPP2A-B5) as part of protein serine/threonine phosphatase complex (GO:0008287), in the category of cellular component. Further, CsPP2A-B1 and CsPP2A-B4 were observed to play a role in regulation of protein phosphatase 2A activity (Molecular function-GO:0008601) (Rossio et al., 2013; McCright & Virshup, 1995). CsPP2A-B4 and CsPP2A-B5 also seemed to be involved in signal transduction (Biological process-GO:0007165)/signaling (Biological process-GO:0023052) (Ruiz-Gómez & Mayor, 1997; Kido et al., 2002; Zhang et al., 2013), and protein phosphatase regulator activity (Molecular function-GO:0019888) (Rossio et al., 2013; McCright & Virshup, 1995). Furthermore, CsPP2A-B9 showed a probable activity in biological processes such as preprophase band assembly (GO:0000913), microtubule cytoskeleton organization (GO:0000226) (Subramanian et al., 2010), and cortical cytoskeleton organization (GO:0030865). In addition, this gene was observed to be part of four cellular components: nucleus (GO:0005634) (Baskaran & Rao, 1991), phragmoplast (GO:0009524), centrosome (GO:0005813) (Jakobsen et al., 2011) and spindle (GO:0005819) (Özlü et al., 2010), alongside showing a probable molecular function in calcium ion binding (GO:0005509)/metal ion binding (GO:0046872) (Bumba et al., 2016; Gugnoni et al., 2017; Brown, 2005). CsPP2A-B7 was also involved in metal ion binding (molecular function-GO:0046872) (Gugnoni et al., 2017; Brown, 2005). CsPP2A-B3 is probably part of HAUS complex (cellular component-GO:0070652) (Lawo et al., 2009), and was seen to have a role in spindle assembly (biological process-GO:0051225) (Matsuoka et al., 2007). As we know, certain PP2A proteins can be associated with microtubular organization and dynamics (Janssens & Goris, 2001; Camilleri et al., 2002; Yoon, Ahn & Pai, 2018; Vavrdová, Samaj & Komis, 2019). The Gene Ontology analysis in this study reveals the role of CsPP2A-B proteins in processes such as microtubule cytoskeleton organization, preprophase band assembly, and spindle assembly, thereby suggesting the involvement of those respective proteins in the above biological processes. Similar functions of PP2A genes can be observed in Arabidopsis, where the B subunit genes play roles in male gametophyte development, ABA signaling, and the regulation of microtubule (MT) activity in both mitotic and non-mitotic cells, among other functions (Máthé et al., 2023).

Figure 6 Gene Ontology (GO) analysis of PP2A genes in C. sinensis.

The GO terms are grouped into three categories, namely biological process, cellular component, and molecular function. The x-axis represents the probable functions whereas the y-axis denotes the frequency of genes part of the functions.

To further comprehend the interaction of CsPP2A proteins with other proteins, a functional interaction network was obtained and analyzed using the STRING database (Fig. 7). The STRING network is based on experimental determination. Due to absence of tea database in the STRING server, the interactome was constructed using the Arabidopsis counterparts (AtPP2AB55β-AT1G17720, AtPP2AB′ζ-AT3G54930 and AtPP2AB′κ-AT5G25510) of tea PP2As involved in the phosphatase activity i.e., CsPP2AB55α-TEA015525.1, CsPP2AB′γ-TEA015045.1 and CsPP2AB′δ-TEA009324.1 respectively, identified through Gene Ontology (GO) analysis. These Arabidopsis proteins are isoforms of the regulatory subunit B, that might be involved in modulation of substrate selectivity and catalytic activity alongside assisting in directing the localization of the catalytic enzyme to a particular subcellular compartment. The STRING network helps to understand the functional interactions among various CsPP2A proteins based on the homologous counterparts (selected based on high bit score) of AtPP2As in tea. From the analysis it was observed that three proteins AtPP2A55Bβ, AtPP2AB′ζ, AtPP2AB′κ interact with PP2A subunit proteins namely PP2AA1 (TEA002042.1), PP2AA2 (TEA002042.1), PP2AA3 (TEA002042.1) along with T5L19.180 (TEA017583.1), F4JC80_ARATH (TEA014397.1), T21H19.130 (TEA029694.1), Q9LSH4_ARATH (TEA003756.1), F4J6D7_ARATH (TEA003756.1), MOB1B (TEA011068.1) and MOB2B (TEA011068.1). TEA002042.1 (CsPP2A-A1) is involved in seedling and floral development, TEA017583.1 (protein phosphatase methylesterase 1) facilitates hydrolase activity, acting on ester bonds (Abdelkafi et al., 2009), TEA014397.1 (RNA polymerase II-associated factor 1) is involved in RNA polymerase II activity (Chen et al., 2009), TEA029694.1 (nucleoprotein TPR) is involved in cell division (Lee et al., 2008), TEA003756.1 (mannosylglycoprotein endo-beta-mannosidase) is an endoglycosidase that hydrolyzes a Manβ1-4GlcNAc linkage in the trimannosyl core structure of N-glycans (Ishimizu et al., 2004), and TEA011068.1 (MOB kinase activator 1) is also in mitosis (Luca & Winey, 1998). The PP2A proteins mainly interact with proteins related to cell division, which indicates their potential role in cell cycle regulation in tea plant. Owing to the cellular functions facilitated by CsPP2A_TON2/FASS1, identified through Gene Ontology (GO) analysis it’s functional-interaction network was generated via Arabidopsis homolog protein (AtPP2A_TON2/FASS1- AT5G18580). CsPP2A_TON2/FASS1 observed to be interacting mainly with PP2A_C (catalytic) subunit proteins namely TEA019439.1 (serine/threonine-protein phosphatase 2A catalytic subunit), TEA015726.1 (serine/threonine-protein phosphatase 2A catalytic subunit), TEA019439.1 (serine/threonine-protein phosphatase 2A catalytic subunit) which facilitate phosphatase activity as well as CsPP2A-A1 (TEA002042.1) (Fig. S3). PP2A_TON2/FASS1 interaction pattern in tea is consistent with PP2A_TON2/FASS1 interaction pattern in Arabidopsis (Spinner et al., 2013).

Figure 7 Functional interaction networks of CsPP2A proteins.

The interaction network was constructed using the Arabidopsis counterparts involved in phosphatase activity (found through GO analysis). Pink lines depicted experimentally validated protein-protein interactions.

Structural analysis of tea PP2A proteins

In order to analyze the structural similarity among PP2A proteins of tea and Arabidopsis plants, superimposition of proteins from both plant species was carried out. The protein structures were superimposed on the basis of BLAST scores and phylogenetic analyses, carried out for the respective plants (Table S1) (Fig. 1). The structural integrity of the predicted proteins was evaluated on the basis of their z-scores obtained from ProSA-web server (Table S5). The z-scores of the predicted protein structures of Arabidopsis ranged from −14.65 to −6.93, whereas those of tea plant were between −12.49 and −3.35. A negative z-score denotes good-quality protein structure (Patra et al., 2019a; Sayeed et al., 2014). PyMOL was used for the superimposition/alignment study of the 3D structures and their structural similarity was measured in terms of root mean square deviation (RMSD) scores (Table S4). The lower the RMSD score of the structures, greater is the similarity between them (Reva, Finkelstein & Skolnick, 1998). Protein models with lowest RMSD scores are displayed in Fig. 8. From subunit A, proteins CsPP2A-A1 and CsPP2A-A2 displayed highest structural similarity with AtPP2A-A2 and AtPP2A-A3 respectively (Table S4) (Fig. 8). From subunit B, protein CsPP2A-B1 exhibited the highest structural similarity with AtPP2A-B55B (AtPP2A-B55β), proteins CsPP2A-B2 and CsPP2A-B3 were aligned to AtPP2A-B′2 (AtPP2A-B′β) to the greatest extent. Similarly, protein CsPP2A-B4 was aligned to AtPP2A-B′8 (AtPP2A-B′θ), CsPP2A-B5 and CsPP2A-B6 were aligned to AtPP2A-B′9 (AtPP2A-B′κ). Furthermore, proteins CsPP2A-B8 and CsPP2A-B9 exhibited highest structural alignment with AtPP2AB′11 (AtPP2AB′β) and AtPP2A-TON2/FASS1. The protein CsPP2A-B7 exhibited sequential similarity with AtPP2A-B′ subfamily in BLAST, phylogeny, intron/exon structural comparison, similarity index analyses as is evident from Figs. 1–3. However, CsPP2A-B7 protein although sequentially similar to AtPP2A proteins, did not align to any of the AtPP2A protein structures (Table S4) exhibiting structural diversity. The structural diversity indicates that CsPP2A-B7 might facilitate other physiological functions in tea plant apart from abiotic stress responsiveness (Pearson, 2013).

Figure 8 The figure depicts PP2A protein structures of Arabidopsis (AtPP2A) and tea (CsPP2A) exhibiting maximum structural similarity.

AtPP2A proteins are depicted in blue whereas CsPP2A proteins are depicted in green.

Expression profiles of tea PP2A genes using transcriptome data

The expression analysis was carried out for the 11 identified CsPP2A genes in response to cold, drought, and salt stress alongside tissue-specific expression, and hormonal treatment. The expression data in the TPIA database is based on the previously conducted experimental studies in response to various abiotic stresses as explained further in the analyses. Moreover, the analysis has been carried out only for the PP2A genes in tea and the analogy is put forward with respect to the expression levels of these genes when exposed to the control condition of each abiotic stress/tissue-specific expression/hormonal treatment. The expression levels were evaluated using transcripts per million (TPM) (Xia et al., 2019).

Tissue-specific expression

CsPP2A genes exhibited varying levels of expression in all the eight tissues. Some genes showed high levels of expression whereas expression level was medium in the other genes (Fig. 9A). In six of these tissues, TEA018948.1 (CsPP2A-B9) was expressed maximally which means that this gene is really important in growth, development and other physiological processes in plant. However, in flower and old leaf, TEA015525.1 (CsPP2A-B1) showed highest expression. Other genes following the highest expressed genes included TEA009324.1 (CsPP2A-B5) (apical bud, young leaf, stem), TEA002042.1 (CsPP2A-A1) (flower, fruit, mature leaf, root, stem), and TEA018948.1 (CsPP2A-B9) (old leaf) (Fig. 9B).

Figure 9 Tissue-specific expression patterns of CsPP2As in 8 different plant tissues.

The y-axis denotes the level of expression in transcript per million (TPM) while the x-axis represents the genes of PP2A-A and B subfamilies (A). Expression patterns of these genes were generated using heatmapper online server and thus depicted as a heatmap (B). The color bar on the top left in (B) represents the normalized TPM values. Green indicates upregulation and red indicates downregulation. The intensity of the colors varies depending on the values. Black represents no expression.

Response to abiotic stresses (cold, drought, and salt stress)

Among all the tea PP2As, TEA018948.1 (CsPP2A-B9) expressed maximally while TEA021728.1 (CsPP2A-B8) expressed minimally throughout all the stages of cold acclimated conditions (Fig. 10A). Exposure to fully acclimated cold condition revealed significant upregulation in 7 out of 11 CsPP2A genes, downregulation in two genes, and negligible change in the remaining two genes. Further under CA 1-7d treatment, all the genes were upregulated except TEA019300.1 (CsPP2A-B3) with negligible change. On decreasing the temperature and thus subjecting to CA 2-7d treatment, majority of the tea PP2A genes (nine genes) showed increased expression levels whereas the remaining two genes were downregulated. When the expression level under recovery condition was analyzed, 10 genes were significantly upregulated whereas one gene (TEA018948.1) (CsPP2A-B9) showed slight increase in its expression (Fig. 10B).

Figure 10 Expression levels of CsPP2A genes under cold stress.

The y-axis denotes the level of expression in transcript per million (TPM) while the x-axis represents the genes of PP2A-A and B subfamilies (A). Expression patterns of these genes were generated using heatmapper online server and thus depicted as a heatmap (B). The color bar on the top left in (B) represents the normalized TPM values. Green color indicates upregulation and red color indicates downregulation. The intensity of the colors varies depending on the values. Black represents no expression.

Under PEG treatment, TEA018948.1 (CsPP2A-B9) exhibited maximum expression whereas TEA021728.1 (CsPP2A-B8) showed least expression levels (Fig. 11A). With increasing exposure time, expression level was consistently decreasing for four CsPP2A genes. In the 24 h period of drought stress condition, eight genes were downregulated, two genes were upregulated, and one gene did not show any significant change. Increase in the time period to 48 h caused the downregulation of six genes and upregulation of five genes. Further prolonged exposure (72 h) led to decreased expression levels in nine genes and increased expression levels in two genes (TEA011483.1 and TEA015525.1) (CsPP2A-A2 & CsPP2A-B1) (Fig. 11B). These two genes were upregulated in all the stages of drought stress treatment and hence seem to play a significant role in drought tolerance alongside the maximally expressed TEA018948.1 (CsPP2A-B9).

Figure 11 Expression levels of CsPP2A genes under drought conditions.

The y-axis denotes the level of expression in transcript per million (TPM) while the x-axis represents the genes of PP2A-A and B subfamilies (A). Expression patterns of these genes were generated using heatmapper online server and thus depicted as a heatmap (B). The color bar on the top left in (B) represents the normalized TPM values. Green color indicates upregulation and red color indicates downregulation. The intensity of the colors varies depending on the values. Black represents no expression.

Under salt stress, TEA018948.1 (CsPP2A-B9) showed maximal levels of expression whereas TEA021728.1 (CsPP2A-B8) showed minimum levels of expression (Fig. 12A). When subjected to NaCl-N-24h condition, 9 out of 11 genes were downregulated while two genes were upregulated. On extending the exposure time to 48 h, nine genes exhibited increased levels of expression while decrease in expression levels was observed in two genes (downregulated previously in the 24 h exposure). Further increment in the time period (72 h) caused downregulation of nine genes and upregulation of two genes (Fig. 12B). It can be observed that none of the genes showed consistent increase or decrease in expression levels in increase in exposure time. Therefore, it can be inferred that different CsPP2A genes might play a role in salt stress tolerance depending on the exposure period.

Figure 12 Expression levels of CsPP2A genes under salt stress.

The y-axis denotes the level of expression in transcript per million (TPM) while the x-axis represents the genes of PP2A-A and B subfamilies (A). Expression patterns of these genes were generated using heatmapper online server and thus depicted as a heatmap (B). The color bar on the top left in (B) represents the normalized TPM values. Green color indicates upregulation and red color indicates downregulation. The intensity of the color varies depending on the values. Black represents no expression.

Response to MeJA (methyl jasmonate) treatment

MeJA treatment for 12 and 24 h showed decreased expression levels in all the genes. However, further prolonged treatment with MeJA (48 h) caused upregulation in three genes out of which TEA018948.1 (CsPP2A-B9) was expressed maximally, followed by TEA002042.1 (CsPP2A-A1). In contrast, TEA021355.1 (CsPP2A-B2) showed minimum levels of expression (Fig. 13).

Figure 13 Expression levels of CsPP2A genes under Methyl-jasmonate (MeJA) treatment.

The y-axis denotes the level of expression in transcript per million (TPM) while the x-axis represents the genes of PP2A-A and B subfamilies (A). Expression patterns of these genes were generated using heatmapper online server and thus depicted as a heatmap (B). The color bar on the top left in (B) represents the normalized TPM values. Green indicates upregulation and red indicates downregulation. The intensity of the colors varies depending on the values. Black represents no expression.

Quantitative real time polymerase chain reaction analysis of tea PP2A genes

Expression patterns of 3 PP2A genes from the B subfamily namely CsPP2A-B2, CsPP2A-B8 and CsPP2A-B9 were analysed through quantitative real time PCR and their relative expression levels under various stress conditions with respect to control condition were calculated using 2−ΔΔCt formula. It was found that CsPP2A-B9/CsPP2A-TON2/FASS1 (TEA018948.1) was upregulated in all three abiotic stress conditions namely, drought stress, salt stress and cold stress as well as hormone treatment. CsPP2A-B2/CsPP2AB’α (TEA021355.1) showed downregulation after 48 h of MeJA treatment and CsPP2A-B8/CsPP2AB″β (TEA021728.1) was observed to be downregulated after 72 h of drought stress, 72 h of salt stress, and 7 days of cold stress (Fig. 14). The results suggest the potential role of (TEA018948.1) CsPP2A-B9/CsPP2A-TON2/FASS1 gene in abiotic stress namely drought, salt and cold stress. The relative expression levels of three PP2A-B subfamily genes are listed in Table S12.

Figure 14 Relative expression of CsPP2A genes under abiotic stresses and hormonal treatment.

Graphs depicting the upregulation and downregulation of CsPP2A genes under various abiotic stresses and hormonal treatment.

Discussion

Genome-wide identification studies help in providing useful information and lay out a preliminary framework for elucidation of functions. The foremost step is to identify the set of gene members in the particular plant of interest and further correlate the structural and functional aspects alongside the expression analysis. We have identified 11 members of the PP2A gene family in C. sinensis. PP2A is a ubiquitous serine/threonine phosphatase involved in various regulatory aspects alongside playing a role in many signaling processes as a secondary messenger (Janssens & Goris, 2001). This gene family in plants have many important physiological roles, among which hormone signaling and stress response are crucial functions of the gene family (Yu et al., 2003; Lillo et al., 2014). The 11 PP2A gene family members in C. sinensis were classified into different subfamilies namely PP2A-A, PP2A-B55, PP2A-B′, PP2A-B″ and PP2A-TON2 through various analyses namely, phylogenetic, intron-exon architecture, similarity index, and domain analyses (Figs. 1–4). Additionally, cis-regulatory elements analysis, gene ontology analysis as well as protein functional interactions analysis provided insights into the potential roles of PP2A proteins in C. sinensis.

Phytohormones such as abscisic acid, auxin, gibberellin, salicylic acid and jasmonic acid play pivotal roles in plant development and physiology (Finkelstein & Zeevaart, 1994; Grill & Himmelbach, 1998; Koornneef et al., 1998; Finkelstein, Gampala & Rock, 2002; Kwak et al., 2002; Das et al., 2025). ABA mediates stomatal closing through increase of cytosolic Ca2+ ions, maintains seed dormancy, and facilitates seed maturation and vegetative growth. ABA signal transduction is positively and negatively regulated by plant PP2As (Kwak et al., 2002; Schmidt et al., 1995; Esser, Liao & Schroeder, 1997; Grabov et al., 1997; Hey et al., 1997; Pei et al., 1997; Wu et al., 1997; Das et al., 2025). In our analysis, CsPP2A CAREs displayed maximum frequency for the function “ABA-responsiveness” (Fig. 5). CAREs are essential for determination of gene transcription, function of genes, and their regulation (Wu et al., 2019; Liu et al., 2021). Moreover, the genes associated with “ABA-responsiveness” (CsPP2A-A1, A2, CsPP2A-B1, B3, B4, B6, B9) showed expression in all the 8 tissues, with CsPP2A-A1 (TEA002042.1) showing highest levels of expression in flower, fruit, mature leaf, root, and stem (Table S6). CsPP2A-A2 and CsPP2A_-B1 were upregulated throughout all stages of drought stress treatment, and CsPP2A_B9 showed maximum level of expression (Table S8). These results suggest CsPP2A gene family as one of the possible regulators of ABA signaling in tea plant for stomatal closing through increase of cytosolic Ca2+ ions, as well as regulators for other ABA functions such as plant growth and development. Through previous research in Arabidopsis, the A subfamily of PP2A gene family was experimentally identified as positive regulator of ABA signal transduction (Kwak et al., 2002).

Gibberellins and salicylic acid are associated with growth and development of the plants by improving the plant’s response to stress (both biotic and abiotic). Both the hormones help the plant withstand or adapt to various stresses though hormonal regulations (Yang et al., 2008; Daviere & Achard, 2015; Gao et al., 2017; Hassoon & Abduljabbar, 2020; Castro-Camba et al., 2022; Das et al., 2025). The CsPP2A genes which contain “Gibberellin-responsive element” (CsPP2A-B5, CsPP2A-B3, CsPP2A-A2) also contain the CAREs for “salicylic acid responsiveness” function (CsPP2A-B5, CsPP2A-B3, CsPP2A-A2, CsPP2A-B6) alongside one extra gene. All these genes are expressed in the abiotic stresses (cold, drought and salt stress) applied in expression analysis for the CsPP2A gene family (Tables S7–S9). These results indicate a possible involvement of these gibberellin and salicylic acid response genes in the regulation during stress conditions experienced by the tea plant. Furthermore, it is observed that in Arabidopsis, the PP2A genes are influenced by gibberellic acid and salicylic acid, indicating their role in responses during plant growth and development (Tan et al., 2020; Kim et al., 2002).

Methyl jasmonates and their derivatives are crucial for plants as they confer defense against herbivore attack and pathogens, and also impart tolerance against abiotic stresses, including ozone, ultraviolet radiation, high temperatures and freezing. They furthermore control various aspects of development such as root growth, stamen development, flowering, and leaf senescence (Howe & Jander, 2008; Wasternack & Hause, 2013; Goossens et al., 2016; Kawahara et al., 2013; Das et al., 2025). CsPP2A genes (CsPP2A-B1, B3, B4, B5, B6, B8, B9, CsPP2A-A1, A2) comprised CAREs for “MeJA responsiveness”. Additionally, the expression pattern analysis of the tea genes in response to MeJA treatment showed that 48 h of MeJA treatment caused upregulation in three genes, among which CsPP2A-B9 indicated maximum level of expression followed by CsPP2A-A1 (Table S10). These results indicate a positive regulation of MeJA signaling in tea plant facilitated through PP2A genes. Similarly, in Arabidopsis, the MeJA hormone affects RCN1, which is homologous to CsPP2A-A1, during stomatal closure (MacRobbie, 1998; Schroeder et al., 2001; Saito et al., 2008). Thus, CsPP2A-B subfamily genes exhibit abiotic stress-related upregulation with CsPP2A-B9/CsPP2A-TON2/FASS1 being the most upregulated gene. A similar upregulation pattern is observed in PP2A-B (regulatory) subunits of soyabean (Glycine max) and wheat (Triticum aestivum) PP2A gene families, 26 members of PP2A-B″ subfamily in soyabean and one B″ subfamily member (PP2A-B″-γ) in wheat were upregulated during abiotic stress conditions (Liu et al., 2019; Xiong et al., 2022).

CsPP2A-B9/CsPP2A_TON2/FASS1 the key regulatory gene under abiotic stress conditions

In Arabidopsis, AtPP2A-TON2/FASS1 protein plays a crucial role in facilitating the formation of cortical microtubules, preprophase bands, and regulating cell growth and elongation, particularly in root cells (Camilleri et al., 2002). Similarly, the TEA018948.1 (CsPP2A-B9/CsPP2A-TON2/FASS1) protein, a homolog of AtPP2A-TON2/FASS1, is involved in various cellular processes, including preprophase band assembly, microtubule organization, cortical cytoskeleton organization, nuclear function, centrosome and spindle formation, and calcium ion binding (Table S3). Additionally, from the domain analysis, it was observed that CsPP2A-B9/CsPP2A-TON2/FASS1 contains EFh domain, which is a calcium ion binding domain. Analysis of cis-acting elements reveals that CsPP2A-B9/CsPP2A-TON2/FASS1 is responsive to abscisic acid, low temperatures, and methyl jasmonate, and contains a protein binding site (Table S2). Overexpression of CsPP2A-B9/CsPP2A-TON2/FASS1 under stress conditions suggests its enhanced activity helps maintain root growth during drought stress and contributes to abscisic acid-mediated resistance to salt and cold stress. The expression patterns of tea genes obtained through qRT-PCR analysis of tea plant under drought, salt, cold stress conditions as well as MeJA treatment were analysed. It was observed that CsPP2A-B9/CsPP2A-TON2/FASS1 (TEA018948.1) related to abscisic acid responsiveness of tea plant, was upregulated during cold, drought and salt stress as well as MeJA treatment (Table S12). Abscisic acid is an important plant phytohormone involved in cold, drought, salt and other abiotic stress related plant defense (Vishwakarma et al., 2017). Furthermore, CsPP2A-B9/CsPP2A-TON2/FASS1 protein is observed to interact with CsPP2A_C (catalytic) subunit proteins as well as CsPP2A-A1, which indicates its potential role in facilitating phosphatase activity in tea (Fig. S3). This signifies that CsPP2A-B9/CsPP2A-TON2/FASS1 (TEA018948.1) is a crucial gene of CsPP2A-B subfamily, regulating plant’s response during abiotic stress conditions.

Conclusion

Plant protein phosphatases are essential regulatory enzymes that play a crucial role in the maintenance of cellular homeostasis and plant growth and development. They also play a critical role in abiotic stress responses such as drought, salinity, and extreme temperatures. PP2A, a type of plant protein phosphatase, has been identified in tea plants through genome-wide analysis and correlating with other in silico analyses, it is believed to facilitate various physiological processes in tea plants. The expression of PP2A genes was observed to increase under biotic and abiotic stress conditions, suggesting a potential positive role of PP2A in stress tolerance in tea plants. PP2A-B subfamily gene CsPP2A-B9/CsPP2A-TON2/FASS1 is the key regulator of abiotic stress response in tea plant as is evident from qRT-PCR analysis. As a crucial regulator of cellular processes, expanding this study of PP2As in tea and in other plants can deepen our understanding of their functions and regulatory mechanisms and pave the way for developing novel strategies for crop improvement and sustainable agriculture.

Supplemental Information

Supplemental Information 1 Supplementary Tables.

Supplemental Information 2 Supplementary Figures.

Supplemental Information 3 MIQE checklist.

The authors would like to thank St. Joseph’s University for providing the facilities required to conduct this research.

Additional Information and Declarations

Competing Interests

The authors declare no conflict of interest.

Author Contributions

Surjit Bhattacharjee conceived and designed the experiments, performed the experiments, analyzed the data, prepared figures and/or tables, and approved the final draft.

Abhirup Paul conceived and designed the experiments, performed the experiments, analyzed the data, prepared figures and/or tables, and approved the final draft.

Aradhana Jana analyzed the data, authored or reviewed drafts of the article, and approved the final draft.

G. Meher Unnati analyzed the data, authored or reviewed drafts of the article, and approved the final draft.

Deepak R. analyzed the data, prepared figures and/or tables, and approved the final draft.

Ye Miao analyzed the data, prepared figures and/or tables, and approved the final draft.

Honglin Lu performed the experiments, analyzed the data, prepared figures and/or tables, and approved the final draft.

Guoxin Shen analyzed the data, authored or reviewed drafts of the article, and approved the final draft.

Neelam Mishra analyzed the data, authored or reviewed drafts of the article, and approved the final draft.

Data Availability

The following information was supplied regarding data availability:

The raw measurements are available in the Supplemental Files.

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
