# Peer review of "Genome-wide identification of PP2A gene family in Camellia sinensis reveals the potential role of CsPP2A-TON2/FASS1 in abiotic stress"

_PeerJ, doi:10.7717/peerj.19431_

## Round 0.1 · original submission · Major Revisions

Please address concerns of all reviewers and amend manuscript accordingly

Reviewer 1 ·

Basic reporting

The manuscript titled "Comprehensive Genome-Wide Analysis of Protein Phosphatase-2A Genes in Tea: Unveiling Their Synergistic Roles in Stress Adaptation" explores the Protein Phosphatase-2A (PP2A) gene family in tea (Camellia sinensis). The study systematically identifies and characterizes these genes using bioinformatics analyses, covering physicochemical properties, chromosomal localization, gene structure, conserved motifs, subcellular localization, transmembrane helices, promoter regions, domain analysis, Gene Ontology (GO) annotation, phylogenetic relationships, and network analysis. Additionally, it investigates the expression patterns of these genes under various biotic and abiotic stress conditions and validates the findings through qRT-PCR analysis, confirming the potential role of CsPP2AB9 in abiotic stress responses.
The study is well-designed and well-written, making a significant contribution to understanding the genetic basis of stress resilience in tea, with potential implications for crop improvement programs. However, a minor revision is recommended—gene naming should be aligned with phylogenetic relationships in accordance with the methodology described by Chao et al., 2020 (Genome-wide identification and expression analysis of the phosphatase 2A family in rubber tree, Hevea brasiliensis).

Experimental design

The study is well-designed and well-written,

Validity of the findings

no comment

·

Basic reporting

In this manuscript the authors describe scaffolding and regulatory subunits of PP2A of Camellia sinensis and its involvement in abiotic stress. The data is valuable since it provides novel facts on functioning of PP2A, which is very important in terms of understanding plant stress reactions.

In brief introduction the authors describe primary function of PP-proteins - the reverse of phosphorylation modification of proteins leading to modulation of their activity. Next, the role of these proteins in hormone metabolism and other plant processes is discussed.

Fig 7: It would be of benefit to filter the network in such way that only the physical interactions are presented.
L359-361: Based on the Methods section, you should provide here that the GO analysis was carried out using “text-mining”, or similar. By the way, the text here and in methods lacks a ton of citations from where the GO annotations were obtained. Also, table S3 lacks citations!

Experimental design

In general, this paper lacks reliable methodological approaches. Also, it would be of a great benefit to the paper to add some “micro-conclusions” at the paragraph ends to emphasize why the particular analysis was made, since the data presented in “Results” is very brief and uninformative. Subsequently - these “micro-conclusions” should be removed from “Discussion” - which will make the presented data clear. The Discussion section in a current form is very big and hard to read.

It's unclear why the A. thaliana was used as reference genome while the Camellia sinensis genome is available:
https://www.ncbi.nlm.nih.gov/datasets/genome/?taxon=4442, and even chromosome-level assembly https://www.ncbi.nlm.nih.gov/datasets/genome/GCA_025759985.1/ (https://www.frontiersin.org/journals/plant-science/articles/10.3389/fpls.2022.1004387/full)

L244: I recommend to compare results of multiple target peptide prediction software along with databases of proved protein subcellular localization. The results of a single localization prediction software are not enough.
L288-301: How can it be that you used BLASTp search for more than 50% of amino acid similarities and here are stated proteins with ~12% of similarity?

Validity of the findings

The Discussion section lacks the comparison of authors' result with recent studies, and part of the Discussion could be moved to Intro (L544-550, L 552-564)

Additional comments

Figure 6. Please do not use serif font a this figure

Reviewer 3 ·

Basic reporting

The title is too long, it is difficult to capture the main message at a glimpse.

The abstract is too long with many unnecessary details, such as “g50% identity, e-value 1e-5”, “cold (4°C), drought (25% PEG 6000), and NaCl (200 mM) and qRT-PCR (22——Ct technique, CsACTIN as internal control) ”.

The abstract has many strange terms without explanation of what they are, such as MEGA, PlantCARE.

Writing needs to be thoroughly improved. There are many sentences that are very long and don’t really make sense, for example,

“Correlating the analyses of the cis-elements, gene ontology and expression patterns of CsPP2A genes under various abiotic and biotic stress conditions, along with prior research on PP2A gene family in other plants, there seems to be involvement of PP2A gene family in regulation of phytohormone (Auxin, Salicylic acid, Gibberellin, Abscisic acid, MeJA) responsiveness during stress conditions in C. sinensi ”.

The introduction section should briefly cover the organism C. sinensis.

The introduction section should mention the key findings in the end.

The result section contains too many meaningless text or details overlapped with Methods section, such as

“The molecular weight of the PP2A proteins ranged between 49400
kDa and 143000 kDa, with TEA019300.1 (CsPP2A-B3) (142997.68 kDa) of PP2A-Bí subfamily being the heaviest and TEA018948.1 (CsPP2A-B9) (49407.77 kDa) of PP2A-B-TON2 subfamily being the lightest.”

“The distribution pattern of introns and exons was obtained via GSDS2.0 software
(http://gsds.cbi.pku.edu.cn/).”

Messages that can be fit into a figure should not lengthily repeat in text, for example,

“In A subfamily, Tea PP2A showed high amino
290 acid sequence identities with Arabidopsis PP2A (CsPP2A-A1/AtPP2A-A1= 85.96%, CsPP2A-
291 A1/AtPP2A-A3= 91.23%, CsPP2A-A1/AtPP2A-A2=94.54% CsPP2A-A2/ AtPP2A-A1=84.99%,
292 CsPP2A-A2/ AtPP2A-A2=92.4%, CsPP2A-A2/ AtPP2A-A3=89.28%). On the other hand, the
293 CsPP2A B-subfamily showed high amino acid sequence identities for some genes whereas low
294 amino acid sequence identities for other genes such as CsPP2A-B1 (CsPP2A-B1/AtPP2A-
295 B_B55A= 79.84%, CsPP2A-B1/AtPP2A-B_B55B=84.86%) showed high amino acid sequence
296 identities with Arabidopsis PP2A. CsPP2A-B6 (CsPP2A-B6/ AtPP2A-B_B55A=12.42%,
297 CsPP2A-B6/ AtPP2A-B_B55A=12.34%) had very low amino acid sequence identities with
Manuscript to be reviewed
298 AtPP2A. There were CsPP2A genes which showed low amino acid sequence identity compared
299 to some of AtPP2A genes and high amino acid sequence identities with other AtPP2A genes such
300 as CsPP2A-B9 (CsPP2A-B9/AtPP2A-B_B55A=12.71%, CsPP2A-B9/ AtPP2A-TON2/FASS1
301 =88.21%) (Fig. 3).”

Experimental design

There are multiple predictors for subcellular localization, the authors should justify why they picked CELLO.

Line 114-128, the authors used TPIA database to obtain PP2A sequences, but then they used these sequences to “search for the PP2A genes in tea genome” database. I don’t get it—since they already obtained PP2A sequences from TPIA, why would they again query the sequences against the tea genome database?

Validity of the findings

The manuscript is overwhelmingly long, which makes it very consuming to read. Most importantly, if the key message is that some PP2A genes are involved in abiotic stress (as suggested by the title), why the authors bothered to perform so many analyses that are entirely irrelevant to this findings, such as phylogenetic analysis, structure prediction, intron/exon structure, similarity, etc. A simple differential gene expression can already confirm that main message.

---

## Round 0.2 · accepted · Accept

All issues pointed out by the reviewers were adequately addressed, and the revised manuscript is acceptable now.